# Optimization of microRNA Acquirement from Seminal Plasma and Identification of Diminished Seminal microRNA-34b as Indicator of Low Semen Concentration

**DOI:** 10.3390/ijms21114089

**Published:** 2020-06-08

**Authors:** Michael Eikmans, Jacqueline D. H. Anholts, Laura Blijleven, Tess Meuleman, Els van Beelen, Marie-Louise P. van der Hoorn, Frans H. J. Claas

**Affiliations:** 1Department of Immunohematology, Leiden University Medical Center, 2333 ZA Leiden, The Netherlands; j.d.h.anholts@lumc.nl (J.D.H.A.); laurablijleven@gmail.com (L.B.); e.van_beelen@lumc.nl (E.v.B.); f.h.j.claas@lumc.nl (F.H.J.C.); 2Department of Gynecology and Obstetrics, Radboud University Medical Center, 6525 GA Nijmegen, The Netherlands; m_tess@hotmail.com; 3Department of Gynecology and Obstetrics, Leiden University Medical Center, 2333 ZA Leiden, The Netherlands; m.l.p.van_der_hoorn@lumc.nl

**Keywords:** semen, seminal plasma, microRNA, optimization, oligozoospermia, asthenozoospermia

## Abstract

About 10–15% of couples who want to conceive suffer from subfertility, while in 30% of these cases, a male factor plays a role. Levels of particular microRNAs in seminal plasma, including those involved in spermatogenesis, may serve as an indicative parameter for subfertility. We first optimized a protocol for acquiring microRNAs from seminal plasma. Next, using a test-validation strategy in a male cohort, we aimed to identify microRNAs of which the levels are related to semen motility and concentration. By qPCR, 742 microRNAs were profiled in three normozoospermic samples, three seminal samples with a low semen motility (asthenozoospermia), and three with a low semen concentration (oligozoospermia). MicroRNAs showing significant differences between groups were further validated in a second cohort consisting of 40 samples with normozoospermia (control group), 47 samples with asthenozoospermia, and 19 samples with oligozoospermia (of which 74% also low motility). Highest microRNA yields were obtained with the Biofluids RNA extraction kit, with inclusion of MS2 RNA carrier and proteinase K treatment to the protocol, and when 50 µL of seminal plasma was used as input. Exosome isolation prior to RNA extraction did not lead to enhanced yields. In the test cohort, 236 microRNAs could be detected, of which 54 microRNAs showed a difference between groups. Five microRNAs were analyzed in the validation cohort. MiR-34b-5p levels in the control group were significantly higher compared to the asthenozoospermia group (*p* < 0.05) and compared to the oligozoospermia group (*p* < 0.001). We optimized microRNA acquirement from seminal plasma and identified microRNA levels in relation to semen concentration and motility. As recent human and mouse studies show that the miR-34 family is a marker of low semen concentration and is crucial in spermatogenesis, seminal plasma miR-34b-5p may represent a suitable candidate to study further as a marker of male subfertility.

## 1. Introduction

Spermatogenesis is the development of spermatozoa in males. Semen consists of many biochemical molecules and is produced and released by specific organs of the male genital system, including seminiferous tubules, epididymis, and accessory glands (seminal vesicles, prostate, and urethral glands). Around 10–15% of couples who want to conceive suffer from complications concerning fertility [1]. Male subfertility is responsible for 30% of cases. Different forms of subfertility can be distinguished. In oligozoospermia, the ejaculate contains a low concentration of spermatozoa, whereas azoospermia is a condition where the semen contain no spermatozoa. This latter condition can be distinguished in non-obstructive azoospermia (disturbed production of spermatozoa) and obstructive azoospermia (obstructed reproductive tract). Asthenozoospermia is a condition whereby the sperm has reduced motility.

Quality of semen is determined according to the volume of the ejaculate (‘V value’), number of spermatozoa (concentration, ‘C value’), and the motility of the spermatozoa (‘M value’). The VCM score represents the multiplication of these three parameters with each other. According to the World Health Organization (WHO) [2,3], semen with a VCM score < 10 has a suboptimal quality and semen with a VCM score > 100 is optimal. Semen with suboptimal quality is related to an increased chance of subfertility. In the field of idiopathic male subfertility, there is no agreement on the use of validated markers. Certainly, there is a need for diagnostic biomarkers in the semen to assess success of conceiving and to better determine the achievement rate of assisted reproductive technologies [4]. Measurement of microRNA levels in seminal plasma may serve and help as indication for subfertility [5,6,7], especially in males with a VCM score between 10 and 100. MicroRNAs are non-coding RNA strands of 20 to 25 nucleotides long, which can regulate gene expression [8]. By binding to the 3 prime-untranslated region (3′-UTR) of mRNA molecules, microRNAs are involved in inhibition of mRNA transcription or of protein translation [9]. Previous research has shown that microRNA levels in seminal plasma differ between infertile men and fertile men and/or between normal and aberrant semen samples [5,10,11,12,13,14].

In this study, we optimized a protocol for the acquirement of microRNA yields from seminal plasma samples. By adhering to a test-validation strategy in a male cohort, we identified microRNAs of which the levels are related to semen motility and concentration.

## 2. Results

### 2.1. Optimization of microRNA Acquirement from Seminal Plasma

To verify that after thawing and centrifugation the supernatant of the seminal plasma contained no sperm cells and somatic cells, five seminal plasma samples were processed for RNA extraction and cDNA synthesis. Presence of somatic RNA (CD45, HPRT-1, ACTB), somatic DNA (HCK), and Y-chromosome-specific DNA (HY, DYS-1) was checked on the cDNA (Table 1). Signals were minimal (at least 64-fold lower) in relation to those of microRNAs (Figure 1, “RNA from SP sup”). To further check presence of genomic DNA in the samples, supernatant and pellet of seminal plasma samples were processed for DNA extraction. The genomic DNA markers showed a relatively low signal in the supernatant of centrifuged seminal plasma samples (Figure 1, “DNA from sup”), whereas a relatively high signal was detected in the pellet (Figure 1, “DNA from pellet”). These observations show that contamination of somatic cells, sperm cells, and genomic DNA is minimal after centrifugation of thawed samples.

We next tested two purification methods for acquirement of RNA from seminal fluid. Highest yields were obtained with the Biofluids RNA extraction kit (Figure 2, first panel). This kit was used for further studies. Since exosomes in fluid may be enriched for microRNAs, we tested whether exosome isolation from seminal fluid prior to RNA extraction gave increased microRNA yields. The volume of seminal plasma as input for the RNA extraction was varied from 50 to 200 µL. The optimal volume of seminal plasma as input for the RNA extraction was 50 µL, and a decrease to 25 µL input did not give enhanced microRNA yields (Figure 2, second panel). Exosome isolation did not enhance yields (Figure 2, third panel), so this step was omitted in further experiments. The Biofluids RNA extraction protocol has the option to incorporate additional proteinase K treatment and MS2 addition. The latter is an RNA carrier that may enhance precipitation of microRNAs during the RNA extraction. For further experiments, we incorporated both proteinase K and MS2 treatment in the RNA extraction procedure, as this resulted in relatively high microRNA yields (Figure 2, fourth panel). The volume of RNA as input for cDNA synthesis was varied from 1 to 4 µL. Highest microRNA yields by qPCR were observed with 2 µL of RNA as input in the cDNA reaction (Figure 2, fifth panel).

### 2.2. Identification of microRNAs Related to Semen Quality: Test Phase

Nine seminal plasma samples were selected that had a total volume between 3 and 6 mL (Figure 3A). For microRNA profiling, we analyzed three normozoospermic samples (VCM > 100; green dots), three samples with asthenozoospermia (<30% motility; orange dots), and three with oligozoospermia (<15 × 10^6^ cells/mL; red dots). The latter two groups had a VCM score < 10. The V, C, M, and VCM scores are depicted in Figure 3A. Levels of 742 microRNAs were determined in each sample. The distribution of Cq value for 236 microRNAs for each sample, which were considered to be significantly expressed, is depicted in Figure 3B. Kruskal–Wallis tests identified 25 microRNAs showing a difference among the three groups of samples with *p* < 0.1. By Mann–Whitney U testing, 54 microRNAs showed a difference (*p* ≤ 0.05) between any two groups (Table 2). Thirty-four with the highest fold difference were included in a hierarchical clustering analysis (Figure 3C).

### 2.3. Identification of microRNAs Related to Semen Quality: Validation Phase

Five microRNAs from the test phase, namely SNORD38b, miR-34b-5p, miR-204-5p, miR-26a-5p, and miR-20a-5p were analyzed in the validation cohort. SNORDB38b and miR-34b-5p were investigated since for both these markers in the test set all samples in the normozoospermia group were positive, whereas samples in one or more of the other groups showed a signal below the threshold. MiR-204-5p was analyzed, since in the test set, the oligozoospermia group showed significant difference with both the control group and the asthenozoospermia group. MiR-26a-5p and miR-20a-5p were studied in the validation set since their levels in the test set were different between the oligozoospermia and asthenozoospermia group and they were highly expressed (average Cq values of 27.1 and 28.8, respectively). After quality control, 33 samples with normozoospermia (N, green dots), 44 samples with asthenozoospermia (AS, orange dots), and 16 samples with oligozoospermia (OL, red dots) were included for further analysis. Since in the oligozoospermia group, 74% also had asthenozoospermia, this group was additionally split into two subgroups for comparison: one with oligozoospermia only (OL only, *n* = 5) and one with both oligozoospermia and asthenozoospermia (OL+AS, *n* = 11).

The differences observed for miR-34b-5p in the test cohort could be verified in the validation cohort (Figure 3D): levels in the normozoospermia group were significantly higher compared to the AS group (*p* < 0.05) and compared to the OL group (*p* < 0.001). In addition, when the OL group was split up into OL only and OL+AS subgroups, the normozoospermia group was significantly higher. In the whole cohort, miR-34b-5p levels significantly correlated with semen concentration (r = 0.36, *p* < 0.0005) and motility (r = 0.28, *p* < 0.01). SNORD38b, miR-20a-5p (both Figure 3D), and miR-204-5p (not shown) were not different between the groups. SNORD38B did show a significant difference between the normozoospermia and OL+AS group (*p* < 0.05). In contrast to its high expression in the test cohort, miR-26a-5p could not be detected in 66% of samples of the validation cohort and was therefore excluded from further analysis. MiR-19b and miR-429, which have been identified to be related to semen quality in previous studies [5,10,11], did not significantly differ in their level between controls and the OL group. Only when the OL group was split up, the samples with OL only (having normal semen motility) had significantly decreased levels of miR-19b-3p compared to the control group (Figure 3D).

## 3. Discussion

In this study, we have optimized a protocol for microRNA assessment in seminal plasma, and we have shown in seminal plasma samples of a cohort of men that levels of microRNAs are related to semen quality. Particularly, we found that levels of miR-34b are diminished in plasma of semen with a low concentration of spermatozoa.

The highest microRNA yields from seminal plasma were obtained when RNA isolation was carried out with spin columns that are particularly appropriate to extract the RNA from biofluids. A previous study where microRNAs were extracted from blood plasma using various RNA extraction procedures showed that the usage of a Biofluids kit gave superior results compared to the other approaches (including a serum/plasma kit and a procedure using TRIzol) [15]. We found that the optimal volume of seminal fluid was 50 µL and the optimal volume of RNA for cDNA synthesis was 2 µL. When using an input of 100–200 µL of seminal plasma for the RNA extraction and/or 4 µL of RNA as input in the cDNA reaction, lower signals were observed in the PCR. This may be due to the presence of inhibitory factors that are present in the seminal plasma. Indeed, a previous study of extracellular microRNA extraction from blood plasma showed an optimum with 100 µL input, whereby an input of 200 µL of plasma led to decreased microRNA yields [15]. The inhibitory effect observed with an excess of volume was ascribed to the protein richness of the blood plasma, which is a characteristic that may be applicable to seminal plasma as well. It is recommended by the company providing the Biofluid kits to apply an RNA carrier to ensure robust RNA isolation. Indeed, we observed that addition of MS2 phage RNA carrier during the procedures did result in increased microRNA yields. Likewise, with RNA extraction from samples of other fluid sources incorporation of an RNA carrier led to higher microRNA yields [15,16,17]. The majority of microRNAs in serum and saliva is concentrated in exosomes [18], and at least part of the microRNA pool in seminal plasma might be present in exosomes [19]. However, in the current study, we found that exosome isolation prior to RNA extraction did not lead to higher PCR yields. This observation corresponds with that from a previous study, showing that cell-free microRNAs in seminal plasma are attached to protein complexes rather than that they are present in exosomes [20]. The conclusion of this previous study may also explain why we acquired higher microRNA yields when including a proteinase K treatment during the RNA extraction procedure. Another study showed that microRNAs constituted about 20% of the total RNA content of exosomes in seminal fluid, but that only a few microRNA species accounted for the majority of the exosomal microRNA content [19].

Despite the availability of reference values by the WHO for semen quality, outcome of semen analysis has a limited prognostic value with respect to predicting the chance of pregnancy: even with a relatively low quality of semen there is still a 40% chance of pregnancy [21]. A Danish study showed that couples, where the male had lower than 20 × 10^6^ spermatozoa per mL in the semen (around the WHO threshold level), had a two-fold reduced chance to become pregnant [22]. The same study showed that a semen motility of less than 30% (WHO threshold level) was associated with a reduced chance to conceive, but in multivariate analysis, semen motility did not have an added prognostic value to semen concentration and morphology [22]. A goal in clinical practice would be to incorporate suitable molecular seminal plasma markers in a model to predict pregnancy success. Moreover, the possibility of measuring molecular analytes in seminal plasma represents a non-invasive approach to determine male subfertility. The WHO defines that semen with a VCM score < 10 has a suboptimal quality and semen with a VCM score >100 is optimal, but the predictive value of samples having an intermediate VCM score may be limited and additional microRNA assessment may be an asset in those cases.

We reckon that extracellular microRNAs represent ideal biomarkers since they are easily accessible, are remarkably stable, can be sampled in a non-invasive manner, and can be reproducibly assessed by qPCR. In a previous study, we demonstrated that microRNAs can even be reliably quantified in conditions of moderate to severe degradation of RNA [23]. As far as the preferred method of detection is concerned when profiling microRNAs in biofluids, the application of qPCR is the recommended approach for sensitive and reproducible quantitative measurements [24]. MicroRNAs represent small non-coding RNA molecules that can bind to the 3′-UTR region of mRNA molecules and thereby negatively affect gene expression. In this way, microRNAs may have an influence on the immune system [25,26]. During conception and pregnancy, seminal plasma may have immune modulating effects on the maternal immune system [27,28,29]. Likewise, microRNAs can regulate the immune system [30,31] and they play a role in spermatogenesis [32,33,34]. A comprehensive and systematic review on the role of microRNAs in male human reproduction was recently published [14].

We adapted a test-validation strategy to identify microRNAs of which the levels are related to semen concentration and motility. To standardize input between samples in the test cohort, microRNAs in each sample were corrected for the median value of its microRNA content. This approach is comparable to global median normalization, as is used for microarray data analysis [35]. It has further been shown that the use of global expression values offers the best strategy for normalization, as long as large numbers of microRNAs are profiled [36]. Since a consensus on reference microRNAs in biofluids is lacking, the choice of standardization in the validation cohort was more difficult. The usage of small nuclear and nucleolar RNAs for normalization is discouraged, since they are longer RNA species than microRNAs and therefore may be purified differently during extraction procedures (miRCURY LNA microRNA PCR instruction manual for serum/plasma and other biofluid samples). Moreover, plasma may not contain nuclear and nucleolar RNAs (https://www.gene-quantification.de). Since particular microRNAs in biofluid that were assumed to serve as an internal control are altered during disease [37], the selection of reference microRNAs should be made based on the most stably expressed markers specific to the sample set one is working with [38]. We therefore selected two microRNAs that showed minimal variation among the samples in the test cohort, and used their levels to normalize levels of microRNAs of interest in the validation cohort.

We found that levels of miR-34b-5p in semen samples with oligozoospermia are decreased compared to those in samples with normozoospermia. Of note in our test cohort, miR-34b-3p and miR-34c-3p, which are family members of miR-34b-5p, showed the same trend. Interestingly, a recent study that incorporated microRNA profiling of testicular biopsies showed that decreased miR-34-5p levels represent a predictive marker of non-obstructive azoospermia (NOA) [39]. Another study also showed that decreased levels of miR-34b and miR-34c, together with two other microRNAs, in testicular tissue, distinguished individuals with NOA and subfertility from controls [40]. Likewise, analysis of spermatozoa showed decreased miR-34b levels in subfertile men [41] and a positive correlation of miR-34c levels with success to conceive [42]. In seminal fluid, miR-34c-5p was one of seven microRNAs for which the levels were decreased in NOA compared to controls [11]. In mice, a deficiency in the miR-34 family leads to impairment of meiosis and maturation of spermatozoa [43], and miR-34-family double-knockout males display infertility due to spermatogenic disruptions [44], suggesting that this family of microRNAs plays an essential role in spermatogenesis.

Both miR-19b and miR-429 showed increased levels in seminal plasma of men with NOA compared to normozoospermia samples [5,10,11]. We could not confirm these findings in our cohort of oligozoospermia samples. Possibly significant differences are only picked up in azoospermia samples and not in oligozoospermia samples, as was described [5].

A strength of our study design is the optimization of acquirement of microRNA yields from seminal plasma samples, the inclusion of a test-validation strategy, and the usage of microRNA panels to screen all known human microRNAs in the test cohort. The sample size in the test cohort was limited, but the applied test-validation strategy is known to decrease the chance to find false positive markers. In the current study, we analyzed one sample from each individual. Hence, we have not investigated intra-individual variability. One limitation of our study is that the semen quality of the samples studied could not be irrefutably linked to male fertility and eventual success to conceive, since this information was not available. Moreover, since samples were collected anonymously, information on the pathological conditions of the males at the time of collection was not available. It may be expected that, next to semen concentration and motility, second level parameters including sperm DNA fragmentation are needed to improve predictive ability with respect to success of conceiving. The extent of sperm DNA damage may even show a relationship to levels of seminal plasma microRNAs, as found in a previous study [45]. To validate findings from the current study, we propose to incorporate in novel experiments the underlying cause of subfertility and the success to conceive, along with biochemical and molecular characteristics of the semen and seminal fluid.

In summary, the microRNAs of interest from the current study, most notably miR-34b, represent suitable candidates in the semen to study further as potential markers of male subfertility.

## 4. Materials and Methods

### 4.1. Study Cohort and Processing of Samples

Semen samples were obtained from men visiting the fertility clinic at the Leiden University Medical Center (LUMC), who had not been able to conceive with the same partner for at least one year. Semen samples were collected anonymously from the fertility clinic. The underlying cause was not documented. Samples were anonymized for the study. Apart from the VCM score, no medical information was known. Semen samples were produced by masturbation, and the subsequent procedure has been described previously [29]. Sperm quality (volume, concentration, motility) was assessed the same day.

The methodology for processing and freezing of samples was carried out following an earlier described protocol [46]: within four hours after collection, semen samples were centrifuged at 2000 rpm for 10 min, sperm cells were discarded, and aliquots of seminal plasma were stored at −80 °C until usage. For molecular assessments, samples were thawed at room temperature and centrifuged at 13,000 rpm for 5 min, and the supernatant was processed for RNA extraction.

To verify that the supernatant of the seminal plasma contained no sperm cells and somatic cells, five seminal plasma samples were processed for RNA extraction following the procedure described in this Methods section. Complementary DNA was transcribed for microRNAs (see below) or from mRNA (following the protocol described in [29]). Somatic RNA markers (CD45, HPRT-1, ACTB) were analyzed by qPCR on cDNA transcribed from mRNAs. In the cDNA synthesis assays, an RNA-minus control was taken along, whereby no RNA was added to the reaction, and whereby markers of somatic DNA (HCK, chromosome 20) and Y-chromosome-specific DNA (HY, DYS-1) were analyzed by qPCR. To further check presence of genomic DNA in the samples, 250 µL of seminal plasma supernatant and the pellet were processed for DNA extraction (QIAamp DNA blood mini kit #51104, Qiagen, Venlo, The Netherlands), and the genomic DNA markers were analyzed by PCR. Primer sequences for all targets are shown in Table 1.

### 4.2. Optimization of RNA Extraction from Seminal Plasmas

Optimization of the RNA extraction procedure was tested on four seminal plasma samples. For isolation of total RNA from seminal plasma samples, two procedures for extraction were compared: the miRCURY RNA isolation kit—Biofluids (#300112; Exiqon, Vedbaek, Denmark) and a combination of TRIzol (Invitrogen, Carlsbad, CA, USA) followed by Nucleospin columns (#740955; Macherey-Nagel, Düren, Germany). Protocols, including the incorporation of proteinase K and MS2 bacteriophage RNA (#10165948001; Roche) to the Biofluid kit method, were carried out according to the supplier’s manuals and a previous paper [16]. Proteinase K was added to the seminal plasma in a final concentration of 2 µg/µL and incubated for 10 min at 37 °C. The input volume of seminal plasma in the RNA extraction kits was tested between 25, 50, 100, and 200 µL. To test whether exosome isolation prior to RNA extraction leads to higher microRNA yields, 50–100 µL of seminal plasma was processed by the miRCURY exosome isolation kit—serum/plasma (#300101, Exiqon, Vedbaek, Denmark). The RNA obtained from each extraction procedure was dissolved in 50 µL of RNAse-free water.

### 4.3. Complementary DNA Synthesis and PCR Amplification

After RNA extraction, reverse transcription reactions were performed following the protocol of the miRCURY LNA guidelines. First-strand synthesis was carried out using the Universal cDNA synthesis kit II (#203301, Exiqon, Vedbaek, Denmark). The reactions (total volume 20 µL) contained 5x reaction buffer, 2 µL of enzyme mix, 1 µL of synthetic RNA spike-in, total RNA template, and nuclease-free water. The amount of RNA input volume in the cDNA synthesis procedure was tested using 1, 2, and 4 µL. The reactions were incubated for 60 min at 42 °C and the reverse transcriptase was inactivated for 5 min at 95 °C.

Following the protocol from Exiqon, PCR reactions (total volume 10 µL) contained 5 µL of PCR master mix, 1 µL of primer mix, and 4 µL of cDNA mix. As a readout for the amount of RNA/cDNA obtained, qPCR was applied for miR-191-5p, miR-10b-5p, miR-195-5p, and miR-103a-3p, since they are present in plasma (Exiqon, guidelines to microRNA profiling in biofluids). Primers for these microRNAs were obtained from Exiqon (#204063, #204753, #339306, and #204306, respectively). PCR assays in 96-wells format or smaller were performed on a MyiQ real-time PCR machine (BioRad, Hercules, CA, USA). PCR assays in 384-wells format were performed on a CFX384 real-time PCR instrument (BioRad, Hercules, CA, USA). The PCR program consisted of 10 min at 95 °C, followed by 45 cycles of 15 s at 95 °C and 1 min at 60 °C. Upon completion of each run, a melting curve analysis was done to check specificity of the primers. The Cq value served as an indication of the amount signal obtained and thus of the efficiency of the extraction procedure.

### 4.4. Identification of microRNAs Related to Semen Characteristics: Test Phase

During collection of the materials, sperm analysis was performed according to the WHO 1999 guidelines [2], based on advice of the European Society of Human Reproduction and Embryology (ESHRE) [47]. Thresholds for determining normozoospermia (VCM > 100), oligozoospermia (<15 × 10^6^ cells/mL), and asthenozoospermia (< 30% motility) were used according to those from the WHO. The level of 742 microRNAs was screened in nine seminal plasma samples, using microRNA ready-to-use PCR human panels (version 3, #203611, Exiqon). This PCR system consists of two 384-wells PCR plates, whereby each well contains a locked nucleic acid (LNA)-containing primer set that targets a particular microRNA. Correction for pipetting variations among different plates was done using the mean Cq values of interplate correction (IPC) wells that are present in triplicate on each plate. MicroRNAs showing a Cq < 35 in at least three of the nine samples were considered to be present, leading to inclusion of 236 microRNA for further analysis. Standardization for input and cDNA synthesis efficiency for each microRNA in each sample was done using the median Cq value of the 236 microRNAs profiled in that particular sample. For this, delta-delta Cq calculation was performed.

### 4.5. Identification of microRNAs Related to Semen Characteristics: Validation Phase

MicroRNAs of interest from the test phase were analyzed in a validation cohort. Based on the findings for two microRNAs (miR-34-5p, miR-20a-5p) from the test cohort, we calculated (https://clincalc.com/stats/samplesize.aspx) that the required sample size is at least 16 per group in the validation cohort, with α = 0.05 and power = 0.80. We included 40 samples with normozoospermia, 47 samples with asthenozoospermia, and 19 samples with oligozoospermia. As standardization for each of the microRNAs of interest, signals of miR-200b-3p and miR-151a-3p were measured. These two microRNAs were selected on basis of their low variation coefficient among the nine samples in the test cohort and the fact that they did not show any differences between the three groups. In a quality check disregarding samples with relatively low microRNA yield, 12 samples showing a significant signal for one reference microRNA (Cq < 35), but not for the other reference microRNA (Cq = 40), were excluded from further analysis. In the validation cohort, the signals of the two reference microRNAs highly correlated (r = 0.89, *p* < 0.00001) with each other. The geometric mean of the two signals (Cq value) was used in the delta-delta Cq formula to correct for the microRNA signals.

### 4.6. Statistics

To analyze differences in microRNA levels between patient cohorts, expression signals were log-transformed. Differences among the three groups included in the test cohort were analyzed by Mann–Whitney U tests (non-parametric test for one group vs. the other) and a Kruskal–Wallis test (non-parametric test for multiple, independent groups). Differences between groups in the validation cohort were analyzed by independent-samples T tests. Analyses were performed in SPSS Statistics version 25 (IBM). For the study where microRNA acquirement from seminal plasma samples was optimized, no statistical tests were applied, but rather decisions for an optimal protocol were made on the basis of differences between the conditions tested.

## Figures and Tables

**Figure 1 ijms-21-04089-f001:**
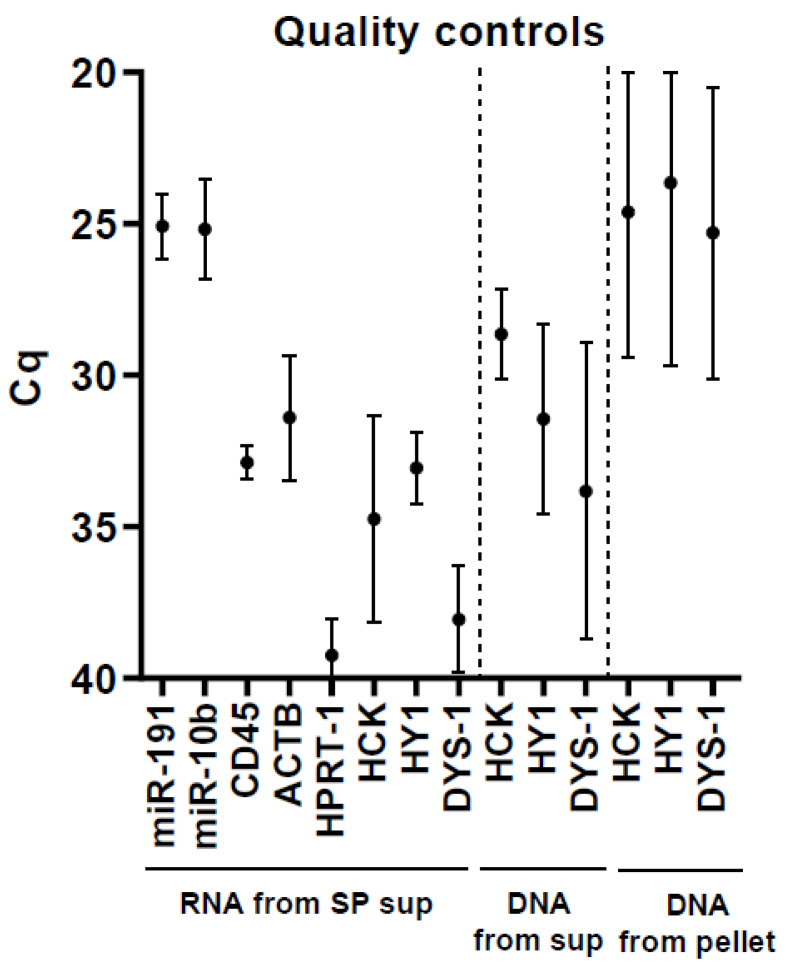
Quality controls to check for contamination in seminal plasma supernatant. To check if the supernatant of the seminal plasma contains sperm cells, somatic cells, and genomic DNA, five seminal plasma samples were processed for RNA extraction and DNA extraction. MiR-191-5b and miR-10b-5p were measured by qPCR on cDNA transcribed for microRNAs. Messenger RNA of CD45, HPRT-1, and ACTB was measured by qPCR on cDNA transcribed for mRNAs. Markers of somatic DNA (HCK) and Y-chromosome-specific DNA (HY, DYS-1) were assessed on cDNA minus-RNA control samples. The genomic markers were also analyzed on DNA obtained from seminal plasma supernatant (“DNA from sup”) and seminal plasma pellet (“DNA from pellet”). Data are represented as means ± SD.

**Figure 2 ijms-21-04089-f002:**
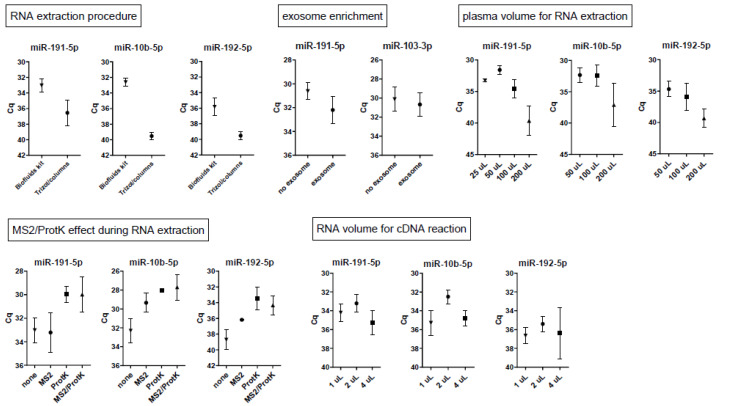
Optimization of microRNA acquirement from seminal plasma. MicroRNA yields were compared by qPCR between different RNA purification methods, with variation of the input volume of seminal plasma in the extraction procedure (25, 50, 100, 200 µL), with or without addition of an exosome enrichment kit, with or without incorporation of MS2 RNA carrier and proteinase K for the RNA extraction procedure, and with variation of the input volume of RNA in the cDNA reaction (1, 2, 4 µL). Data are represented as means ± SEM.

**Figure 3 ijms-21-04089-f003:**
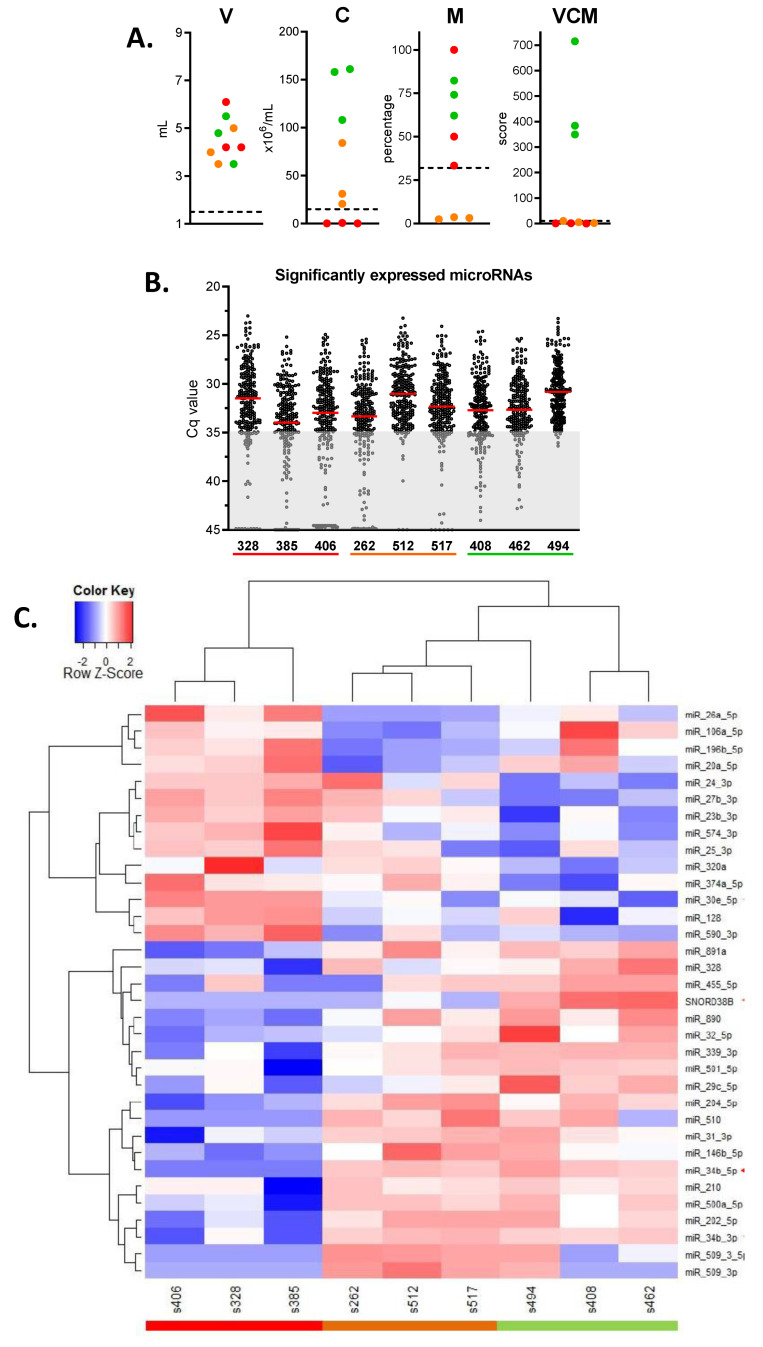
Identification of microRNAs related to semen quality. (**A**) Nine seminal plasma samples were selected with a volume between 3 and 6 mL: three normozoospermic samples (VCM > 100, green dots), three samples with asthenozoospermia (<30% motility; orange dots), and three with oligozoospermia (<15 × 10^6^ cells/mL; red dots). (**B**) The distribution of Cq value for 236 microRNAs, which were considered to be significantly expressed in the test cohort. Red lines represent the median value in each sample. (**C**) Hierarchical clustering analysis of 34 microRNAs that showed the biggest difference between groups. (**D**) Representation of levels of microRNAs of interest in individual patient samples from both the test cohort and the validation cohort. Boxes represent medians with interquartile range. N, normozoospermia; AS, asthenozoospermia; OL, oligozoospermia; OL+AS, oligoasthenozoospermia. *, *p* < 0.05. #, *p* < 0.001.

**Table 1 ijms-21-04089-t001:** Primer sequences of mRNA- and genomic DNA targets ^1^.

	Forward Primer	Reverse Primer
CD45	5^^′^^-GGTCGTCAAACAAAAACTTCCC-3^^′^^	5^^′^^-TGAGATCCATCCCTGCAGTG-3_^′^_
HPRT-1	5^^′^^-AGATGGTCAAGGTCGCAAGC-3^′^	5^′^-TCAAGGGCATATCCTACAACAAAC-3^′^
ACTB	5^′^-ACCACACCTTCTACAATGAG-3^′^	5^′^-TAGCACAGCCTGGATAGC-3^′^
HCK	5^′^-TATTAGCACCATCCATAGGAGGCTT-3^′^	5^′^-CTTCCGCTCCACTTTCCCTAAC-3^′^
HY	5^′^-TGGCGATTAAGTCAAATTCGC-3^′^	5^′^-CCCCCTAGTACCCTGACAATGTATT-3^′^
DYS-1	5^′^-TCCTGCTTATCCAAATTCACCAT-3^′^	5^′^-ACTTCCCTCTGACATTACCTGATAATTG-3^′^

^1^ Primers for CD45, hypoxanthine-guanine phosphoribosyltransferase-1 (HPRT-1), and beta-actin (ACTB) are directed against mRNA. Primers for HCK, HY, and DYS-1 are directed against genomic DNA.

**Table 2 ijms-21-04089-t002:** MicroRNAs showing a significant difference between groups in the test phase ^1^.

	N vs. Low-C	N vs. Low-M	Low-C vs. Low-M	K-W	Average Cq
miR-23b-3p	x	x	-	x	28.6
miR-27b-3p	x	x	-	x	28.9
miR-99b-5p	x	x	-	x	29.8
miR-152	x	x	-	x	30.8
SNORD38b ^2^	x	x	-	x	33.7
miR-204-5p	x	-	x	x	31.7
miR-221-3p	x	-	x	x	32.1
miR-34b-5p ^2^	x	-	x	x	32.3
miR-146b-5p	x	-	x	x	32.6
miR-510	x	-	x	x	37.9
miR-100-5p	-	x	x	x	31.1
miR-509-3p	-	x	x	x	39.9
miR-891a	x	-	-	-	27.2
miR-888-5p	x	-	-	-	28.2
miR-25-3p	x	-	-	-	29.5
miR-574-3p	x	-	-	x	30.3
miR-374b-5p	x	-	-	-	30.3
miR-30d-5p	x	-	-	-	31.4
miR-598	x	-	-	-	33.6
miR-32-5p	x	-	-	x	33.6
miR-16-1-3p	x	-	-	x	33.9
miR-501-5p	x	-	-	-	34.1
miR-590-3p	x	-	-	-	34.7
miR-339-3p	x	-	-	x	35.2
miR-34c-3p	x	-	-	-	36.7
miR-99a-5p	-	x	-	-	26.4
miR-106a-5p	-	x	-	-	27.8
miR-24-3p	-	x	-	-	28.4
let-7d-3p	-	x	-	-	31.6
miR-376c-3p	-	x	-	-	33.9
miR-328	-	x	-	-	34.1
miR-665	-	x	-	x	36.1
miR-142-3p	-	x	-	-	37.0
miR-455-5p	-	x	-	-	37.8
miR-26a-5p	-	-	x	-	27.1
miR-20a-5p	-	-	x	-	28.8
miR-205-5p	-	-	x	-	28.1
miR-222-3p	-	-	x	-	29.4
miR-196b-5p	-	-	x	x	30.5
miR-532-5p	-	-	x	-	32.1
miR-892a	-	-	x	-	32.1
miR-18b-5p	-	-	x	-	33.0
miR-202-5p	-	-	x	x	34.7
miR-455-3p	-	-	x	-	35.3
miR-500a-5p	-	-	x	x	35.3
miR-502-3p	-	-	x	-	35.8
miR-31-3p	-	-	x	-	35.9
miR-181b-5p	-	-	x	x	35.9
miR-34b-3p	-	-	x	x	36.2
miR-202-3p	-	-	x	x	36.8
miR-502-5p	-	-	x	-	37.4
miR-449a	-	-	x	-	37.9
miR-335-3p	-	-	x	x	38.9
miR-509-3-5p	-	-	x	x	39.2

^1^ MicroRNAs marked with x in the first three columns showed a significant difference (*p* ≤ 0.05) between two groups in a Mann–Whitney U test. MicroRNAs marked with x in the fourth column showed the most significant difference (*p* < 0.1) among all groups in a Kruskal–Wallis test. N, normozoospermia samples; OL, oligozoospermia samples; AS, asthenozoospermia samples. Average Cq values of the nine samples in the test set are shown. ^2^ For both SNORD38b and miR-34b-5p, all samples in the normozoospermia group of the test cohort were positive, whereas all samples in one or two of the other groups showed a signal below the threshold. For both, average Cq was calculated for only the samples showing a significant expression.

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
