# Peer review of "Optimization of microRNA Acquirement from Seminal Plasma and Identification of Diminished Seminal microRNA-34b as Indicator of Low Semen Concentration"

_ijms, 2020, doi:10.3390/ijms21114089_

Round 1

Reviewer 1 Report

Comments of the manuscript entitled “Optimization of microRNA acquirement from seminal plasma and identification of diminished seminal microRNA-34b as indicator of low semen concentration” by Michael Eikmans, Jacqueline D.H. Anholts, Laura L. Blijleven, Tess Meuleman, Marie-Louise P. van der Hoorn and Frans H.J. Claas.

The present study aimed to optimize a protocol for acquiring microRNAs from seminal plasma and subsequently to identify microRNAs of which the levels are related to semen motility and concentration. The authors concluded that the miR-34 family is a marker of low semen concentration and is crucial in spermatogenesis, seminal plasma miR-34b-5p may represent a suitable candidate to study further as marker of male subfertility.

The manuscript contains interesting basic data and is well organized. However, some specific issues need to be addressed before publication:

  1. Please include a sample size justification in Material and Methods part.
  2. Include a detailed frozen method paragraph.
  3. My bigger concern is about quality controls. The authors should prove that the results are only from seminal plasma with no contamination with genomic DNA, somatic cells, sperm cells, or exosomes. I suggest the inclusion of some makers: Individual PCRs to check the absence of genomic DNA (PRM1, PRM2 and/or GAPDH genes), absence of somatic RNA (CD45 gene), etc.
  4. In the validation part, the authors compare the results between 1) normo (control) vs. astheno, and 2) normo vs. oligo. However, in the oligo population, 74% has also low motility. Please include a new comparison between 3) normo vs. isolated oligo and 4) normo vs. oligoastheno.
  5. Compare, in the discussion part of the article, the present results with studies in sperm samples and exosomes samples. Moreover, the discussion should mention the last comprehensive systematic review: 10.1111/ANDR.12714
  6. Include 2 clear paragraphs at the end of the discussion part:
    1. Strengths and limitations.
    2. Propose new experiments to confirm the findings.

Author Response

Reviewer 1

The present study aimed to optimize a protocol for acquiring microRNAs from seminal plasma and subsequently to identify microRNAs of which the levels are related to semen motility and concentration. The authors concluded that the miR-34 family is a marker of low semen concentration and is crucial in spermatogenesis, seminal plasma miR-34b-5p may represent a suitable candidate to study further as marker of male subfertility.

The manuscript contains interesting basic data and is well organized. However, some specific issues need to be addressed before publication:

  1. Please include a sample size justification in Material and Methods part.

We have added the sample size calculation to the Methods section (see page 13, lines 347-350).

  1. Include a detailed frozen method paragraph.

Much of the methodology on processing and freezing samples has been described in of our earlier studies (Meuleman et al, J Reprod Immunol 2015). We have included this reference and the accompanying information on this topic as a separate paragraph in the revised draft of the manuscript (see page 11, second paragraph).

  1. My bigger concern is about quality controls. The authors should prove that the results are only from seminal plasma with no contamination with genomic DNA, somatic cells, sperm cells, or exosomes. I suggest the inclusion of some makers: Individual PCRs to check the absence of genomic DNA (PRM1, PRM2 and/or GAPDH genes), absence of somatic RNA (CD45 gene), etc.

Thank you for your comment. To address this, we have carried out additional experiments.
We have checked signals of somatic mRNA (CD45, HPRT-1, beta-actin), somatic DNA (HCK), and Y-chromosome-specific DNA (HY, DYS-1) on cDNA synthesized from extracted RNA from supernatant of seminal plasma samples. Also, DNA was extracted from the seminal plasma to find out if genomic DNA could be detected by PCR. Contamination of genomic DNA, somatic cells, and sperm cells is minimal in the supernatants after centrifugation of thawed samples. The results of these experiments have been included in the revised draft of the manuscript (see page 4, first paragraph and Figure 1).

  1. In the validation part, the authors compare the results between 1) normo (control) vs. astheno, and 2) normo vs. oligo. However, in the oligo population, 74% has also low motility. Please include a new comparison between 3) normo vs. isolated oligo and 4) normo vs. oligoastheno.

We have additionally split up the oligozoospermia group into oligo only and oligo+astheno, and have made comparisons accordingly (see page 5, last paragraph and Figure 3d).

  1. Compare, in the discussion part of the article, the present results with studies in sperm samples and exosomes samples. Moreover, the discussion should mention the last comprehensive systematic review: 10.1111/ANDR.12714

Thank you for your useful comment. We have added the reference by Salas-Hueto et al. where the role of microRNAs in human male reproduction was investigated by systematic review. We also have included additional comparisons of our findings with those from others and have included these in the Discussion section (see page 8, lines 218-220 and page 9, second and third paragraph).

  1. Include 2 clear paragraphs at the end of the discussion part:
    1. Strengths and limitations.
    2. Propose new experiments to confirm the findings.

Thank you for these suggestions. We have added this information to the Discussion section (see page 9, last paragraph and page 10

Reviewer 2 Report

I read with interest this study which I consider innovative. The objective of the study is clear and the translational repercussions seem rather evident. However there are some points that deserve improvement.

Introduction

Specify better that in the field of idiopathic male infertility there is no agreement on the use of validated markers.

Methods

Specify that the spermiogram is performed according to the WHO 2010 rules (5th edition) and how many operators are involved.

The major criticality of the study, presented strangely as a strength by the authors, lies in the lack of information relating to the patient selection criteria. Do these markers suffer from different pathological conditions observable at the time of collection? (e.g. does male hypogonadism have the same relevance as varicocele or urogenital infection?).

Another relevant aspect concerns the possible intra-individual variability of these markers. Was the test repeated on the same individual? how many times, after how many days?

Another important aspect, what are the authors' goals for clinical practice? Can the use of these markers affect the pregnancy rate?

The number of cases is limited, but all in all, this aspect can be considered and accepted in the limitation paragraph.

Finally, I believe there is a basic problem. The authors consider the traditional seminal parameters (density, motility, morphology) closely correlated with the fertilizing capacity. In reality we know that second level parameters are needed to improve this predictive ability (e.g. analysis of sperm DNA fragmentation). Have the authors examined correlations with this aspect?

Author Response

Reviewer 2

I read with interest this study which I consider innovative. The objective of the study is clear and the translational repercussions seem rather evident. However there are some points that deserve improvement.

Introduction

Specify better that in the field of idiopathic male infertility there is no agreement on the use of validated markers.

Thank you for your comment. We have added additional information on this topic in the Introduction section (see page 3, second paragraph).

Methods

Specify that the spermiogram is performed according to the WHO 2010 rules (5th edition) and how many operators are involved.

The fertility center at our hospital center used the WHO 1999 classification for semen analysis during the collection of sperm samples for this study. In the Netherlands many centers are using the WHO 1999 classifications, mainly based on advices from the ESHRE. Since last year we are using the WHO 2010 criteria, however, not during the collection of the materials for this study. We have added this information to the manuscript (see page 12, lines 332-334).

The major criticality of the study, presented strangely as a strength by the authors, lies in the lack of information relating to the patient selection criteria. Do these markers suffer from different pathological conditions observable at the time of collection? (e.g. does male hypogonadism have the same relevance as varicocele or urogenital infection?).

Thank you for this comment. Since samples were collected anonymously, information concerning the underlying cause was not available. We have added this information to the manuscript (see page 9, last paragraph). 

Another relevant aspect concerns the possible intra-individual variability of these markers. Was the test repeated on the same individual? how many times, after how many days?

We were not able to study intra-individual variability since one sample per individual was available. We have added this information to the manuscript (see page 9, last paragraph).

Another important aspect, what are the authors' goals for clinical practice? Can the use of these markers affect the pregnancy rate?

Yes, that it would indeed be the goal that these markers aid in enhancing predictive value with respect to pregnancy success. We have included this information in the Discussion section (see page 8, lines 201-202).

The number of cases is limited, but all in all, this aspect can be considered and accepted in the limitation paragraph.

Thank you. We have added this information as part of the Discussion section (see page 9, last paragraph).

Finally, I believe there is a basic problem. The authors consider the traditional seminal parameters (density, motility, morphology) closely correlated with the fertilizing capacity. In reality we know that second level parameters are needed to improve this predictive ability (e.g. analysis of sperm DNA fragmentation). Have the authors examined correlations with this aspect?

Unfortunately we did not have available information on DNA fragmentation. We do acknowledge that this parameter may enhance predictive ability and have added this information to the Discussion section (see page 9, last paragraph and page 10, first paragraph).

Round 2

Reviewer 1 Report

All the reviewers' comments have been satisfactorily addressed.

The reviewer does not have more questions/comments.

Reviewer 2 Report

Accepted in the present form